# Impact of Using Smartphone While Walking or Standing: A Study Focused on Age and Cognition

**DOI:** 10.3390/brainsci13070987

**Published:** 2023-06-23

**Authors:** Tayla B. Lino, Gabriella S. Scarmagnan, Sidney A. Sobrinho-Junior, Giovanna M. F. Tessari, Glaucia H. Gonçalves, Hugo M. Pereira, Gustavo Christofoletti

**Affiliations:** 1School of Medicine, Institute of Health, Federal University of Mato Grosso do Sul, UFMS, Campo Grande 79060-900, Brazil; tayla.lino@ufms.br (T.B.L.); simoes.gabriella@gmail.com (G.S.S.); junioorsobrinho@gmail.com (S.A.S.-J.); gioftessari@gmail.com (G.M.F.T.); glaucia.goncalves@ufms.br (G.H.G.); 2Department of Health and Exercise Science, University of Oklahoma, Norman, OU 73019, USA; hugomax@ou.edu

**Keywords:** smartphone, cell phone, pedestrian, multitasking behavior, gait, mobility, balance

## Abstract

Background: Using smartphones during a task that requires upright posture is suggested to be detrimental for the overall motor performance. The aim of this study was to determine the role of age and specific aspects of cognitive function on walking and standing tasks in the presence of smartphone use. Methods: 51 older (36 women) and 50 young (35 women), mean age: 66.5 ± 6.3 and 22.3 ± 1.7 years, respectively, were enrolled in this study. The impact of using a smartphone was assessed during a dynamic (timed up and go, TUG) and a static balance test (performed on a force platform). Multivariate analyses of variance were applied to verify main effects of age, task, estimates of cognitive function and interactions. Results: Compared to young, older individuals exhibited a poorer performance on the dynamic and on the static test (age effect: *p* = 0.001 for both variables). Dual-tasking with a smartphone had a negative impact on both groups (task effect: *p* = 0.001 for both variables). The negative impact, however, was greater in the older group (age × task effect: *p* = 0.001 for both variables). Executive function and verbal fluency partially explained results of the dynamic and static tests, respectively. Conclusions: The negative impact of using a smartphone while performing tasks similar to daily activities is higher in older compared to young people. Subclinical deficits in distinct aspects of cognitive function partially explain the decreased performance when dual-tasking.

## 1. Introduction

Smartphones have become an integral part of people’s daily lives, offering not only the ability to communicate with others but also a wide range of useful features. With all its attractions, the number of mobile network subscriptions worldwide reached almost 6.6 billion in 2022, indicating that a significant portion of the world’s population use a smartphone [1].

The popularity of smartphones is related to the ability to easily carry and allow internet access at a somewhat affordable price, compared with a desktop computer [2]. Many of daily tasks became easier with the introduction of smartphone devices. People can do online shopping, order food, monitor health, perform bank transactions, receive exercise instructions, listen to music, and work [2,3,4,5,6,7]. If on the one hand, smartphones brought advantages in people’s lives, then on the other, they came with potential risks. For example, some individuals may become addicted to using the device [8,9,10].

Dual-tasking with a smartphone has become a common phenomenon in today’s society. It is common to see commuters using their smartphones in bus stops or walking while using their smartphones. This practice, however, may lead to distractions and falls [11]. The performance of daily tasks involving smartphone usage frequently relies on the functional connectivity between motor and cognitive neural networks. When multiple tasks are performed simultaneously, the neural networks experience increased demand, which can lead to conflicts in task prioritization. Engaging in the simultaneous performance of multiple tasks, such as walking and texting or talking on the phone, can place additional demands on brain regions associated with attention, executive function, and planning leading to a potential negative impact in motor performance [11,12,13,14,15,16,17].

Several studies investigated the effects of dual-tasking in young and older individuals [18,19,20]. Fewer studies, however, specifically used smartphones for texting and talking while simultaneously walking [21,22,23,24]. The majority of the studies with smartphones were conducted in a sample of young individuals, perhaps on the assumption that this population use their device more often [25,26]. In the last decade, however, there has been a significant increase in the proportion of older individuals using smartphones [27,28,29]. This can be attributed to a combination of two factors: older individuals are adopting the new technology, as well as individuals who started using smartphones at younger ages, and are now part of the older age group [30,31,32,33].

Given the growing number of older individuals using smartphones, it is important to examine the impact of dual-tasking with their smartphones on daily tasks, such as walking and standing. The increased cognitive demand of using a smartphone while performing a motor task can be particularly challenging for older individuals when compared to their younger counterparts, due to age-related changes in both motor and cognitive function [34,35,36,37,38,39]. In the presence of a cognitive task, the motor performance tends to exhibit greater performance decrements than the cognitive performance during a cognitive-motor dual task [34]. 

In this study, we investigated the effects of texting messages and talking on the phone on subjects’ performance during a static standing task (evaluated with a force platform) and a dynamic task (assessed using the timed up-and-go test, TUG). Our primary hypothesis was that smartphone use would negatively impact the results of the balance test with a greater magnitude in older, rather than young, individuals. We also hypothesized that results of the cognitive tests would partially explain the reduction in performance in the presence of smartphone use.

## 2. Materials and Methods

All evaluations were performed at the Laboratory of Biomechanics and Clinical Neurology of the Federal University of Mato Grosso do Sul. The laboratory is located in an area of approximately 100 ft^2^. The assessments were controlled in terms of floor regularity, lighting (six 9W lights), temperature (between 75 and 85 °F), and disturbing sounds (up to 40 dB). The study was in accordance with the Declaration of Helsinki [40] and the guidelines on good clinical practice [41]. All eligible participants signed the consent form prior to the assessments. The protocol was approved by the institution’s Ethics Committee (# 3,678,458).

The recruitment of participants involved a combination of direct contact with potential subjects and the utilization of social media platforms. Participants were offered reimbursement for expenses related with their transportation to the laboratory. 

Our sample consisted of 51 older (36 women) and 50 young (35 women) (age: 66.5 ± 6.3 vs. 22.3 ± 1.7 years old, respectively). Included individuals had their own smartphone and were able to walk and stand without assistance. The use of the participant’s own smartphone minimized any learning effect on a new device. Exclusion criteria comprised cases of neurological or psychiatric disorders, and use of orthoses or prostheses in lower limbs. Additionally, participants were screened for dementia using multiple cognitive tests, including the Mini-Mental State Examination [42], the Frontal Assessment Battery [43], and the Semantic Verbal Fluency test [44]. Individuals who exhibited scores lower than the cutoff values, adjusted by education and age established for the local population, [45,46,47] were excluded from the study.

Figure 1 details the number of participants included and excluded during recruitment process. One hundred and forty-five subjects were assessed for eligibility. Due to the inclusion and exclusion criteria, the sample size was reduced to 101 participants. Table 1 details individual characteristics including functional independence [48], cognition [42,43,44], and daily use of smartphone.

The methodological procedures are described according to the STROBE Statement checklist [49]. Two motor tasks were used to investigate motor performance during both static and dynamic conditions. Each motor task was performed in the presence and absence (randomized order) of a smartphone.
(1)To assess static upright-postural control, a balance task was performed on a force platform composed of a 500 mm^2^ plate and four load cells (BIOMEC 400_V4^®^, EMG System). Participants performed all tests in their bare feet and were instructed to remain standing on the platform for 60 s. Variables assessed were maximum sway in the anterior-posterior and medial-lateral directions (defined as any movement of the top of a vertical member relative to its bottom and measured in cm), center of pressure sway area (defined as the change in center of pressure over time, and calculated as an elliptical base in cm^2^), and average imbalance speed (defined as the speed deviation of the center of pressure, measured in cm/s). The data were processed using MATLAB^®^ (The Mathworks, Natick, MA, USA). The routine was defined for a sampling of 100 Hz, with a second-order digital low-pass Butterworth filter at 35 Hz. A larger sway in the anterior-posterior and medial-lateral directions, along with a larger center of pressure sway area and faster imbalance speed, are indicative of a poorer postural balance.(2)To investigate motor performance on a dynamic test, we used the TUG test [50]. Results from this test include the time and number of steps necessary to rise from a chair, walk 3 m, return, and sit in the same chair. A longer time to complete the task and a greater number of steps, relative to the control condition (i.e., no smartphone) and the other group (young vs. old), was indicative of compromised dynamic balance. The number of steps was calculated relative to each individual height to account for between subject differences in step length [51].

All motor tasks were performed with and without smartphone use. Each participant was asked to perform the tasks while answering a phone call or texting messages. The researchers ensured that cell phone activities (talking or texting messages) were performed throughout the entire TUG test and force platform assessment. All participants answered the same questions during the smartphone tasks. This study focused on the cost of using a smartphone on the static and dynamic tests (i.e., impairment in TUG performance and force platform results) and not the cognitive cost (i.e., reduction in accuracy of typing or talking on the smartphone). The use of a smartphone was imposed to each motor task, and the safety of participants was prioritized by asking the individuals to maintain their balance during each trial. All participants had their smartphone in their pants front pocket before starting the static or dynamic tests. The repeated measures design used in this study minimized potential differences in the clothing effect between groups and individuals. 

To investigate the role of important aspects of cognitive function on the motor performance, three instruments were used: the Mini-Mental State Examination [42], the Frontal Assessment Battery [43] and the Semantic Verbal Fluency test [44]. The Mini-Mental State Examination assessed the general cognition of the participants, including temporal and spatial orientation, registration of three words, attention and calculation, immediate and delayed recall, language, and visual-constructive practice. The score ranged from 0 to 30 points, with lower scores indicating worse cognitive performance [45]. To quantify executive function, the Frontal Assessment Battery was used. The test provides insights on concept recognition, lexical flexibility, motor programming, conflicting instructions, inhibitory control, and environmental autonomy. The instrument score ranges from 0 to 18 points and lower scores indicate worse cognitive performance [46]. The Semantic Verbal Fluency test registered the number of animals a person could count aloud during 60 s. This test was used to assess lexical knowledge and semantic memory organization. A higher number of animals named by the participants indicates better cognitive scores [47].

### Statistical Analysis

The data are reported as mean ± standard deviation. The statistical hypotheses were tested with the multivariate analyses of variance (MANOVA) in association with Wilk’s Lambda test to verify the main effect of age (older × young), sex (men × women), task (no smartphone × texting × talking on the phone) and interactions. If necessary, separate univariate analysis of variance (ANOVA) were conducted to provide complementary assessments for each factor. Pairwise comparisons were performed using post-hoc tests with Bonferroni correction. Effect sizes (ES, η^2^p) and statistical power are reported [52]. Cognition and hours per day using a smartphone were included as covariates to account for expected age-related changes in cognitive functions and differences in smartphone usage (Table 1) [53]. In all analyses, significance was set 5%.

## 3. Results

Older individuals were slower and took more steps than young participants during the dynamic balance test (main effect of age: *p* = 0.001, ES = 0.277, statistical power of 99.9%). The use of a smartphone increased the number of steps and walking time in both groups (main effect of task: *p* = 0.001, ES = 0.686, statistical power of 99.9%). Dual-tasking with a smartphone had a greater negative impact on older, compared to young, participants, for both the time and number of steps (group × task effect: *p* = 0.001, ES = 0.246, statistical power of 99.6%). Compared with the no smartphone condition, texting messages resulted in greater negative impact in the time and number of steps relative to talking on the phone (Table 2). 

There was no influence of sex or interaction with use of smartphone on the relative number of steps (main effect of sex: *p* = 0.070, ES = 0.034, statistical power of 44.3%; and sex × task effect: *p* = 0.812, ES = 0.002, statistical power of 8.2%). Older women had a greater number of steps than older men and young adults (age × sex effect: *p* = 0.007, ES = 0.072, statistical power of 77.7%). No interaction was seen between age, sex and task (*p* = 0.547, ES = 0.006, statistical power of 15.0%). Additionally, men and women had similar results on the time to complete the dynamic test without interaction with use of a smartphone (main effect of sex: *p* = 0.941, ES = 0.001, statistical power of 5.1%; sex × task effect: *p* = 0.760, ES = 0.003, statistical power of 9.3%; age × sex effect: *p* = 0.680, ES = 0.002, statistical power of 6.9%; and age × sex × task effect: *p* = 0.690, ES = 0.004, power of 10.9%) (Figure 2). 

In Table 3 we present the impact of cognition on each variable of the dynamic balance test, by examining the primary effects of cognition, as well as the interaction effect between cognition and dual task performance in young and older adults. Multivariate analysis identified that the Frontal Assessment Battery accounted for poor results in the dynamic test (*p* = 0.002, ES = 0.161, statistical power of 92.8%). Conversely, there were no significant cognition × task effect for the Mini-Mental (*p* = 0.440, ES = 0.039, statistical power of 29.0%) and for the Semantic Verbal Fluency test (*p* = 0.645, ES = 0.026, statistical power of 19.9%).

Table 4 details the static balance of the participants. The older group had poorer results in the static test than the young group (main effect of age: *p* = 0.001, ES = 0.260, statistical power of 99.3%). Using a smartphone impaired balance in both groups (main effect of task: *p* = 0.001, ES = 0.582, statistical power of 99.9%). Compared to the no smartphone condition, talking on the phone resulted in greater changes in postural stability relative to texting messages. The reduction in balance in the presence of smartphone use was larger in older, compared with young, participants (group × task effect: *p* = 0.001, ES = 0.189, statistical power of 84.1%). 

During the control condition (i.e., without a smartphone), women had a reduced performance in the lateral sway compared to men (sex effect: *p* = 0.015, ES = 0.064; statistical power of 68.8%), center of pressure (*p* = 0.005, ES = 0.084, statistical power of 80.9%), and lateral speed (*p* = 0.022, ES = 0.058, statistical power of 63.3%). However, sex did not influence frontal sway (sex effect: *p* = 0.473, ES = 0.006, statistical power of 11.0%) or frontal speed (sex effect: *p* = 0.078, ES = 0.034, statistical power of 42.0%).

In the presence of a smartphone, women had greater impairments in lateral sway (sex × task effect: *p* = 0.021, ES = 0.042, statistical power of 70.4%), fluctuations in the center of pressure (sex × task effect: *p* = 0.026, ES = 0.040, statistical power of 67.6%), and frontal speed (sex × task effect: *p* = 0.003, ES = 0.063, statistical power of 87.6%), compared with men. There was no significant sex × task effect for frontal sway (*p* = 0.068, ES = 0.030, statistical power of 53.3%) or lateral speed (sex × task effect: *p* = 0.089, ES = 0.027, statistical power of 48.9%).

Older women had greater lateral sway than other groups (sex × age: *p* = 0.008, ES = 0.076, statistical power of 77.1%). There was no significant sex × age effect for frontal sway (*p* = 0.447, ES = 0.007, statistical power of 11.8%), lateral sway (*p* = 0.858, ES = 0.003, statistical power of 8.4%), center of pressure (*p* = 0.099, ES = 0.030, statistical power of 37.7%), and frontal speed (*p* = 0.447, ES = 0.007, statistical power of 11.8%).

The use of a smartphone had similar effects in young and older men and women for frontal sway (sex × task × age effect: *p* = 0.846, ES = 0.022, statistical power of 7.6%), lateral sway (sex × task × age effect: *p* = 0.202, ES = 0.018, statistical power of 33.8%), center of pressure (sex × task × age effect: *p* = 0.263, ES = 0.015, statistical power of 28.7%), frontal speed (sex × task × age effect: *p* = 0.581, ES = 0.006, statistical power of 13.9%), and lateral speed (sex × task × age effect: *p* = 0.623, ES = 0.005, statistical power of 12.7%). Figure 3 details the static balance of the participants, considering the factors sex, age, and task.

Table 5 details the impact of cognition on each variable of the static test. Multivariate analyses identified that the Semantic Verbal Fluency test influenced the static balance results (cognition × task effect: *p* = 0.014, ES = 0.061, power of 92.3%), but the other cognitive test results had no influence on static balance (Mini-Mental cognition × task effect: *p* = 0.963, ES = 0.010, statistical power of 18.9%; Frontal Assessment Battery: *p* = 0.094, ES = 0.045, power of 79.2%).

Older individuals mostly use their smartphones for leisure (43.1%), followed by receiving calls (37.3%), and updates/news (19.6%). Young adults use their smartphones for leisure (62.0%), studying (20.0%), and receiving calls (18.0%). 

Multivariate analysis identified that daily hours of smartphone use partly explained the negative impact on the dynamic test (main effect of the daily hours with smartphone: *p* = 0.016, ES = 0.081, statistical power of 73.5%; daily hours with smartphone × task effect: *p* = 0.015, ES = 0.121, statistical power of 81.7%). Conversely, the hours of smartphone use did not impact the static balance test (main effect of the daily hours with smartphone: *p* = 0.773, ES = 0.028, statistical power of 18.0%; daily hours with smartphone × task effect: *p* = 0.443, ES = 0.111, statistical power of 49.2%) (Table 6).

## 4. Discussion

This study examined the impact of texting messages and talking on the phone on individuals’ performance in a static and a dynamic balance task. In addition, we investigated the role of age and specific aspects of cognitive function on walking and standing tasks in the presence of a smartphone. Key findings are: (1) Using a smartphone simultaneously with a dynamic and static task that requires upright-postural control (i.e., TUG and standing on a force platform, respectively) had a greater negative impact in older, compared to young, adults; (2) Poor results in cognitive tests frequently used to quantify verbal fluency and executive function partly explain the negative results of motor tasks either in the presence or absence of a smartphone; and (3) The reduction in motor performance depends on the type of activity executed on the smartphone. Specifically, texting messages showed a greater negative impact during walking, whereas talking on the phone induced greater negative interference during the static balance test, regardless of age or sex. 

Our experimental protocol carefully required the use of a smartphone during standing and walking because these are common tasks in activities of daily living, such as waiting for a bus, standing in line at a grocery store or walking. The risk for accidents and injury was previously suggested to be enhanced in case of distractions, such as when using smartphones [11]. Findings from the current study provide relevant information about the influence of different aspects of cognitive function on motor tasks. Specifically, for both young and older individuals, poor performance on the Frontal Assessment Battery was specifically associated with a decreased performance on the dynamic test, while poor performance on the Semantic Verbal Fluency test was associated with a decreased performance on the static balance test. These findings corroborate previous observations reporting the impact of cognitive function on motor performance [54,55,56,57,58].

Recent studies have examined the risks associated with using smartphones while walking or standing and support the findings of the current study. For example, during a dynamic task, Bianchini et al. [59] observed a reduced performance of a walking and turning task in both young and older individuals in the presence of a smartphone. Belur et al. [60] reported the magnitude of smartphone interference during a walking task was greater in older individuals and depended on the results of a verbal fluency task. Additionally, during a static upright-postural task, Onofrei et al. [21] showed that young individuals had greater difficulty in maintaining balance while talking on the phone, compared to texting, but the effect of aging was not investigated. What sets our study apart from previous ones is that we systematically quantified the impact of specific cognitive domains on both walking and standing tasks in young and older individuals. Our study also provides additional evidence regarding the specificity of cognitive domains on distinct aspects of motor function. This is in agreement with Yashida et al. [61] and Takeuchi et al. [62] who observed the impact of executive function on the performance of walking while simultaneously using a smartphone, as well as the findings from Rodríguez–Aranda et al. [63] and Makizako et al. [64] showing that verbal fluency had an impact on a finger tapping task. 

In this current study, both groups had greater difficulty in texting messages while walking, compared to the talking while walking task. Explanations of this finding are likely explained by the fact that texting during walking generates visual distraction, causes gait instability, and competes for limited visual resources with other environmental cues [65,66,67,68,69,70,71]. Conversely, talking on the phone had a greater influence on static balance than texting. A potential physiological mechanism explaining the effects of talking on the smartphone while walking is the activation of brain areas associated with upright-postural control and movement, such as the cerebellar-cortical pathways, along with the prefrontal and premotor pathways during a conversation [21,70,71,72,73].

Significant differences between sexes on static and dynamic tasks were observed in this study. Specifically, older women performed the walking test with more steps compared to both older men and the young group. Furthermore, for the static task, women exhibited poor performance in the metrics of lateral sway, center of pressure sway area, and lateral speed when compared to men. These findings were anticipated, considering that certain motor patterns can vary between sexes [74]. However, the negative impact of using a smartphone was similar between the sexes and age groups (i.e., no significant age × sex × task effects) for both static and dynamic tasks.

Daily hours of smartphone usage partially accounted for the negative results observed in the dynamic test, but not in the static test. This task specificity was unexpected and highlights the need for further research investigating the effects of varying daily hours of smartphone usage on motor function. Those investigations would contribute to elucidating the extent to which smartphone usage influences motor abilities and potentially provide insights into strategies for mitigating any negative impacts.

This study examined the motor cost of walking or standing still while simultaneously using a smartphone. Although technology can be used to offload attentional resources (e.g., using navigation maps on the smartphone), it is important to acknowledge the potential cognitive cost and cognitive failures involved. Cognitive failures are cognitive-based errors that can occur during tasks that individuals are typically capable of completing. Factors such as memory, attention, and distractibility can influence the quality of the cognitive task. To obtain a more comprehensive understanding of the implications and potential risks associated with smartphone use, in conjunction with secondary tasks, it is important to further investigate additional factors, such as the frequency of smartphone checking and the duration of smartphone screen time [75]. By considering these factors, we can better assess their impact on the safety of using a smartphone and obtain insights into its usage in various contexts.

### Limitation

The authors recognize some limitations. Firstly, the assessments were performed in a controlled environment (laboratory). The results assessed in public places may be different due to external distractors, such as pedestrians, vehicles flow, uneven floor, noises, and adverse weather conditions. Secondly, the study considered the variations among smartphone brands and models. Different phones can feature a distinct interface for each application, potentially leading to varying impacts on users’ attention and cognitive load. Thirdly, we did not control the potential influence of cultural backgrounds and educational levels on the results. Despite all individuals scoring within the normal range of cognitive tests [42,43,44,45,46,47], it is plausible that cultural background and different levels of education could have positive or negative effects on the outcomes. Fourthly, our study did not evaluate typing speed and grammar errors during texting. It is possible that certain individuals prioritized the motor tasks over the texting task, leading to variations in attention and performance. While this aspect was not specifically measured in our study, it provides opportunities for future research. Finally, older individuals reported lower use of a smartphone per day than young adults. Although this study was conducted during the COVID-19 pandemic and many older individuals used a smartphone to avoid loneliness and depression [76], this was not enough to match smartphone usage time between young and older individuals in our sample.

## 5. Conclusions

The negative impact of using a smartphone while performing walking and standing tasks, similar to the ones required in daily activities, was higher in older, compared to young, individuals. Results of cognitive tests targeting executive function and verbal fluency partially explained the impaired performance when a smartphone was simultaneously used during walking and standing tasks. These findings should not discourage older individuals to use smartphones, but should alert them about the risks involved if the device is used simultaneously with a walking or standing task. Further studies are necessary to investigate potential physiological mechanisms involved. 

## Figures and Tables

**Figure 1 brainsci-13-00987-f001:**
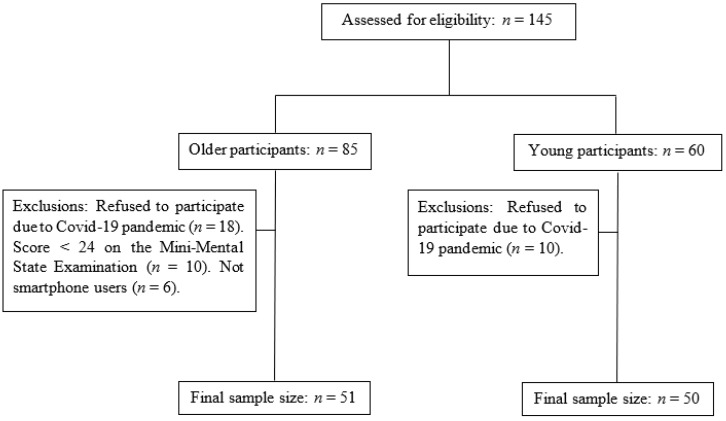
Description of number of participants included in each group.

**Figure 2 brainsci-13-00987-f002:**
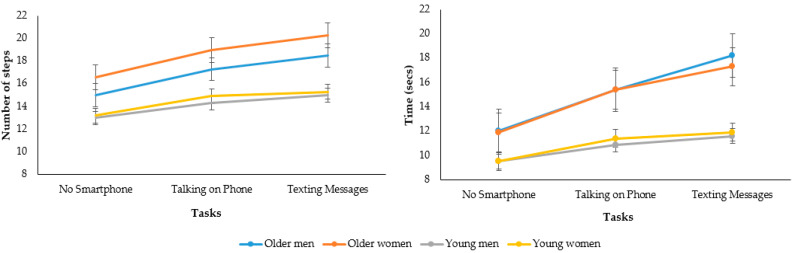
Participants’ performance on the TUG test considering sex, age, and task.

**Figure 3 brainsci-13-00987-f003:**
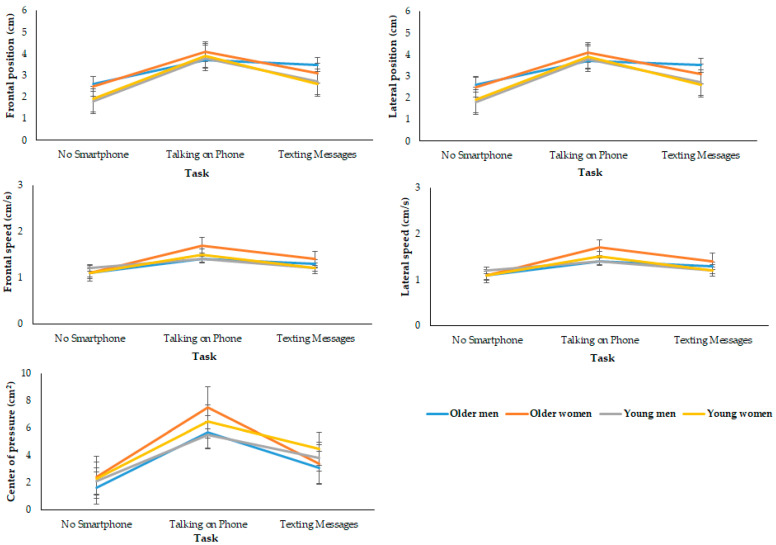
Participants’ performance on the standing test, considering sex, age, and task.

**Table 1 brainsci-13-00987-t001:** General characteristics of the participants.

Variables	Older	Young	95% Confidence Interval	*p*
Sample size, %	50.5	49.5	---	0.921
Sex (female), %	70.6	70.0	---	0.948
Age, years	66.5 ± 6.3	22.3 ± 1.7	42.4 to 46.1	0.001
Functional independence, score	0.5 ± 3.1	0.1 ± 0.3	−0.4 to 1.3	0.289
Mini-Mental State Examination, score	26.7 ± 2.6	28.8 ± 1.2	−2.8 to −1.2	0.001
Frontal Assessment Battery, score	15.0 ± 2.6	17.3 ± 1.0	−3.1 to −1.6	0.001
Verbal Fluency test, *n*. of animals	16.6 ± 4.0	23.0 ± 4.7	−8.1 to −4.7	0.001
Hours per day using smartphone, h	2.9 ± 1.7	5.4 ± 2.7	−3.4 to −1.6	0.001

Data are presented in percentage, absolute number and mean ± standard deviation. *p* value of the chi-square test for the categorical variables and *p* value of the Student *t*-test for the continuous variables.

**Table 2 brainsci-13-00987-t002:** Impact of dual task with a smartphone while walking in young and old persons.

Walking Test	Groups	Tasks			MANOVA Main Effect		
		No Cell Phone	Talking on Phone	Texting Messages	Group	Task	Interaction
Time, secs	Older	12.0 ± 2.9	15.4 ± 5.0 ^Ʝ^	17.6 ± 5.6 ^Ʝ,ᶲ^	*p* = 0.001 ES = 0.277 Power = 99.9%	*p* = 0.001 ES = 0.686 Power = 99.9%	*p* = 0.001 ES = 0.246 Power = 99.6%
	Young	9.5 ± 1.4	11.2 ± 1.9 ^Ʝ^	11.8 ± 2.1 ^Ʝ,ᶲ^			
Step, *n*	Older	16.1 ± 3.4	18.5 ± 3.9 ^Ʝ^	19.8 ± 4.7 ^Ʝ,ᶲ^			
	Young	13.2 ± 1.3	14.8 ± 1.8 ^Ʝ^	15.3 ± 1.8 ^Ʝ,ᶲ^			

Data are expressed in mean ± standard deviation. *p*-value, effect size (ES) and statistical power analyses of the MANOVA tests. Univariate analyses (ANOVA) confirmed group, task and interaction effects on time and number of steps. ^Ʝ^ Post hoc indicated differences compared to the “no smartphone”. ^ᶲ^ Post hoc indicated differences compared to the “talking on the smartphone”.

**Table 3 brainsci-13-00987-t003:** Effect of the covariate factor cognition on the walking test. Data is for young and older groups combined.

Covariate Factor	TUG Variables	Cognition Main Effect			Cognition × Task Main Effect		
		*p*	Effect Size	Power (%)	*p*	Effect Size	Power (%)
Mini-Mental State Examination	Time	0.176	0.019	27.1	0239	0.015	30.6
	Steps	0.571	0.003	8.7	0.609	0.005	13.1
Frontal Assessment Battery	Time	0.001	0.100	88.9	0.001	0.085	97.1
	Steps	0.061	0.036	46.8	0.023	0.038	69.1
Semantic Verbal Fluency test	Time	0.003	0.089	85.9	0.334	0.011	24.2
	Steps	0.030	0.048	58.8	0.462	0.008	18.1

*p*-value, effect size (ES) and statistical power analyses of the ANOVA tests, considering cognition as covariate.

**Table 4 brainsci-13-00987-t004:** Impact of dual task with smartphone on postural balance in young and older individuals.

Static Test	Groups	Tasks			MANOVA Main Effect		
		No Cell Phone	Talking on Phone	Texting Messages	Group	Task	Interaction
Frontal sway, cm	Older	2.5 ± 1.3	4.0 ± 1.9 ^Ʝ^	3.2 ± 1.2 ^Ʝ,ᶲ^	*p* = 0.001 ES = 0.260 Power = 99.3%	*p* = 0.001 ES = 0.582 Power = 99.9%	*p* = 0.001 ES = 0.198 Power = 84.1%
	Young	1.9 ± 0.6	3.9 ± 2.1 ^Ʝ^	2.6 ± 0.9 ^Ʝ,ᶲ^			
Lateral sway, cm	Older	1.4 ± 0.9	2.7 ± 2.1 ^Ʝ^	1.7 ± 0.7 ^Ʝ,ᶲ^			
	Young	1.8 ± 0.5	2.6 ± 0.9 ^Ʝ^	2.3 ± 0.6 ^Ʝ^			
Center or pressure, cm^2^	Older	2.2 ± 1.8	7.1 ± 6.6 ^Ʝ^	3.3 ± 2.3 ^Ʝ,ᶲ^			
	Young	2.3 ± 1.2	6.2 ± 5.5 ^Ʝ^	4.3 ± 4.7 ^Ʝ,ᶲ^			
Frontal speed, cm/s	Older	1.2 ± 0.3	1.7 ± 0.5 ^Ʝ^	1.3 ± 0.3 ^Ʝ,ᶲ^			
	Young	1.1 ± 0.2	1.5 ± 0.4 ^Ʝ^	1.2 ± 0.2 ^Ʝ,ᶲ^			
Lateral speed, cm/s	Older	1.0 ± 0.2	1.4 ± 0.4 ^Ʝ^	1.1 ± 0.3 ^Ʝ,ᶲ^			
	Young	1.1 ± 0.2	1.3 ± 0.2 ^Ʝ^	1.2 ± 0.2 ^Ʝ,ᶲ^			

Data are expressed in mean ± standard deviation. *p*-value, effect size (ES) and statistical power of the MANOVA tests. Univariate analyses (ANOVA) confirmed group effect on frontal sway, task effect on all variables, and interaction effect on lateral speed. ^Ʝ^ Post hoc analyses indicated differences in each group compared to “no smartphone” condition. ^ᶲ^ Post hoc analyses indicated differences in each group compared to “talking on the smartphone” condition.

**Table 5 brainsci-13-00987-t005:** Effect of the covariate factor cognition on the static balance test. Data is for young and older groups combined.

Covariate Factor	Motor Variables	Cognition Main Effect			Cognition × Task Main Effect		
		*p*	Effect Size	Power (%)	*p*	Effect Size	Power (%)
Mini-Mental State Examination	Frontal sway	0.794	0.001	5.8	0.850	0.002	7.5
	Lateral sway	0.901	0.001	5.2	0.775	0.003	9.0
	Center or pressure	0.749	0.001	6.2	0.783	0.003	8.8
	Frontal speed	0.261	0.014	0.201	0.756	0.003	9.4
	Lateral speed	0.320	0.011	16.7	0.919	0.001	6.3
Frontal Assessment Battery	Frontal sway	0.024	0.057	62.5	0.236	0.016	30.9
	Lateral sway	0.072	0.036	43.8	0.767	0.003	9.1
	Center or pressure	0.002	0.100	87.3	0.483	0.008	17.3
	Frontal speed	0.025	0.055	61.3	0.026	0.041	67.8
	Lateral speed	0.087	0.033	40.3	0.781	0.003	8.8
Semantic Verbal Fluency test	Frontal sway	0.008	0.078	77.1	0.233	0.016	31.1
	Lateral sway	0.872	0.001	2.6	0.768	0.003	9.1
	Center or pressure	0.001	0.112	90.9	0.037	0.037	62.7
	Frontal speed	0.539	0.004	9.4	0.004	0.060	85.2
	Lateral speed	0.243	0.015	21.4	0.463	0.006	11.3

*p*-value, effect size (ES) and statistical power of the ANOVA tests, considering cognition as covariate.

**Table 6 brainsci-13-00987-t006:** Effect of the covariate factor hours of using a smartphone on the walking and static tests. Data is for young and older groups combined.

Covariate Factor	Motor Variables	Cognition Main Effect			Cognition × Task Main Effect		
		*p*	Effect Size	Power (%)	*p*	Effect Size	Power (%)
Hours of using smartphone	Steps	0.058	0.036	47.6	0.029	0.036	66.4
	Time	0.913	0.001	5.1	0.345	0.011	23.6
	Frontal sway	0.960	0.001	5.0	0.868	0.002	7.2
	Lateral sway	0.403	0.008	13.2	0.600	0.006	13.4
	Center or pressure	0.652	0.002	7.3	0.793	0.003	8.6
	Frontal speed	0.792	0.001	5.8	0.714	0.004	10.3
	Lateral speed	0.609	0.003	8.0	0.622	0.005	12.7

*p*-value, effect size (ES) and statistical power of the ANOVA tests, considering hours of using a smartphone as covariate.

## Data Availability

The data presented in this study are available on request from the corresponding author.

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
