# Peer review of "Impact of Using Smartphone While Walking or Standing: A Study Focused on Age and Cognition"

_brainsci, 2023, doi:10.3390/brainsci13070987_

Round 1

Reviewer 1 Report (New Reviewer)

While the study raises interesting questions regarding the impact of smartphone use on motor performance in different age groups, there are several limitations and issues that need to be addressed.

Within the experimental group, the middle-aged adults showed a relatively low average MMSE score of 26.7, which was accompanied by significant deficits in multiple cognitive test domains when compared to young adult participants. The observed cognitive profile is relatively uncommon among middle-aged adults; therefore, it is plausible that the participants in this study may not adequately represent a typical sample of middle-aged adults with intact cognitive capacity. This finding may have implications for the reliability of the study results, particularly with respect to middle-aged populations.

Additionally, the study did not consider differences between different smartphone brands and models. Different phones may have different interfaces and applications, which could have varying effects on users’ attention and cognitive load.

Finally, the study did not consider other factors such as cultural background and education level, which may influence the results.

Overall, the quality of English language in this research article appears to be OK.

Author Response

Reviewer 2 Report (New Reviewer)

This is an interesting study on the impact of using smartphone while walking or standing. The paper is well-written and timely. I agree that it will contribute well to the literature. I only have a few comments to improve the manuscript further:

1. In the introduction, the authors highlighted accurately that the practice of using smartphone may lead to distractions and falls. However, recent study has also shown that some types of smartphone usage (e.g., using maps or tools-related apps) may help to offload attentional resources and reduce cognitive failure. I believe that it is worth to mention this in the introduction/discussion to provide a more balanced view on this issue

Smartphone use and daily cognitive failures: A critical examination using a daily diary approach with objective smartphone measures. (2023). British Journal of Psychology, 114(1), 70-85.   2. The methodology of the current study is strong and I appreciate the efforts to conduct the study well. In the Method section, it will be helpful to include information on how the sample was recruited and the compensation.   3. It will be useful to clarify the type of effect size reported in the results section.   4. It might be useful to report a zero-order correlation table for all the main variables in the current study.

Round 2

Reviewer 1 Report (New Reviewer)

Many studies suggest that an MMSE value of less than 27 may indicate the presence of mild cognitive impairment, impaired cognition, or cognitive dysfunction. In the present study, the middle-aged group exhibited a mean MMSE score of less than 27 points, which is highly probable that some participants in this group do not conform to the normal range of cognitive ability. Consequently, the reliability of the research result is questionable due to the comparatively lower mean MMSE score of the middle-aged group in the present study.

Ref:

[1]The relationship between health-related quality of life and higher-level functional capacity in elderly women with mild cognitive impairment.

[2]The association between structure-function relationships and cognitive impairment in elderly glaucoma patients.

[3]Study of the value of homocysteine levels in predicting cognitive dysfunction in patients after acute carbon monoxide poisoning.

[4]Preoperative Cognitive Impairment as a Predictor of Postoperative Outcomes in Elderly Patients Undergoing Spinal Surgery for Degenerative Spinal Disease.

Author Response

Reviewer 2 Report (New Reviewer)

The authors have revised the manuscript well. The paper is ready for publication. Congratulations

Author Response

This manuscript is a resubmission of an earlier submission. The following is a list of the peer review reports and author responses from that submission.

Round 1

Reviewer 1 Report

Dear authors,

This study investigated the performance change of walking and standing during using smartphone, and also tried to explain the reasons of changes in the view of cognition. The study design is good and accurate. The writing of manuscript is clear and logical. Congratulation to authors. Minor suggestions as follows:

Both paragraph 1 and 2 are related with basic introduction of smartphone, better to combine them into one paragraph.

I noticed there are some very short paragraphs in the method section, such as lines 126-128. Actually, lines 126-133 introduced the test of cognitive function, one paragraph is better.

Best wishes

Reviewer 2 Report

General Comments

This study aimed to identify the impact of talking on a phone and texting while walking on the motor components of these dual-task performances. However, the authors appear to have not controlled the cognitive task performance or collected cognitive task performance variables, limiting the author’s ability to identify the true impact on the motor components of the dual tasks. Additionally, the authors did not control for group differences in smartphone use or more specifically assess hours spent using the smartphone for phone calls versus texting. Finally, there are many grammar errors (aka, missing words, shifting tenses, etc.) throughout all sections of the manuscript that will need to be corrected.

Abstract

Page 1, lines 17-18: The age effects results should be written as how the older adults performed relative to the young adults, which is what an age effect says, as opposed to an objective statement about how older adults performed.

Introduction

Page 1, lines 33-34 and 38-39: A few sentences in the second paragraph seem beyond the point of the study, such as the fact that smartphones allow for portable internet access and that individuals can become addicted to smartphone use. In general, that paragraph seems extraneous.

Page 1, line 44: Please clarify what you mean by “potential of greater demand on the integration of these neural networks”. This sentence is currently unclear because the quoted phrase is a bit vague.

Page 1, lines 44-45: The driving sentence does not seem relevant.

Page 2, lines 46-49: These sentences should be rewritten to improve clarity. The “motor performance tends to lose quality” is vaguely stated and the next sentence seems relevant but out of place. More background and details are needed to explain the relevance of increased demand on cognitive resources for dual-task performance.

Page 2, line 67: The introduction supports why performing a TUG test would be relevant, but there is nothing in the introduction that explains why a static balance test is a particularly dangerous time to text or mimics an everyday task that older adults are performing.

Page 2, lines 71-74: You need to include a secondary purpose statement, as opposed to just including the hypothesis, or better identify that the cognitive tests are being assessed as covariates in the primary aim analysis. As currently stated, it is unclear what aim and analysis are being done relative to this secondary hypothesis.

Methods

Page 4, lines 109-110: Please better describe your balance variables. Are the sway values a range (maximum sway in anterior-posterior and medial-lateral directions)? Was the center of pressure area calculated as an ellipse, a square or some other shape, such as a 95% ellipse (covering 95% of the sway area)? And is the imbalance speed an average or maximum velocity value in each anterior-posterior and medial-lateral direction or just the average or maximum velocity measured?

Page 4, lines 120-125: Please include details about how the cognitive task difficulty was controlled with the phone call, so that all participants experienced a similar phone call and had to respond to similar questions. Also, how were texting situations handled in which participants completed the required text message before the end of the TUG trial? The cognitive load would be very different for someone who was a quick texter and able to complete the text before finishing the task versus someone who was a slower texter and did not complete the text before finishing the task. Additionally, how was cognitive performance on the phone call (length and complexity of utterances) and texting (number of errors in the message and potentially time to complete the text if they finished before the end of the motor trial) recorded and considered in terms of dual-task costs? Someone who performed well on the gait and cognitive task would have exhibited a very different dual-task performance than someone who performed well on the gait task but not on the cognitive task, but they would appear to perform the dual-task similarly if you only recorded and analyzed gait/balance performances. Finally, did you record any seated cognitive trials? In order to assess dual-task interference on cognition, you would need to record single-task cognitive trials.

Page 4, lines 147-149: Why were the hours per day using a smartphone not included in the analysis as a covariate? The hours were statistically different between the older and young groups, which likely dramatically affected dual-task performances, beyond any age-related differences. Also, did you collect any information about what types of activities were typically done on the smartphone? Some people may spend more time talking on the phone or texting, which would impact their dual-task performances during the phone call and text message trial conditions. It is not enough to include this difference as a limitation of the study in the Limitations section.

Results

Page 5, lines 182-186 and Table 3. It is very unclear where these results are coming from. The statistical analysis section in the Methods section made it seem like the cognitive tests were being input as covariates into the MANOVA for the balance and TUG variables. However, I only see effect sizes for the 3 cognitive tests during general smart phone use, as opposed to during the three different trial conditions (no smartphone, phone call, text message). Why were the results not reported for Task main effects and separated by task?

Page 7, lines 227-230 and Table 4: Same comments as for lines 182-186 and Table 3.

Discussion

Page 8, lines 233-238: These two sentences muddle the two aims of the study and treat them as though they’re one analysis as opposed to two. Please revise these sentences to better identify what was specifically done in this study.

Page 8, line 241: What does the “with greater magnitude refer to”? I believe that you’re saying worse performance on cognitive tests impacted TUG performance more than standing balance performance, but this sentence needs to be clarified.

Page 8, lines 242-243: It should be clarified that you’re referring to the impact on the motor task during the dual-task performance.

Page 8, lines 251-253: This sentence is misleading and needs to be rewritten. Your study did not assess lower extremity muscle performance but instead assessed static balance and dynamic balance and walking performances that may or may not be related to status of lower extremity muscles.

Page 8, lines 254-256: I don’t believe it is correct to say that the FAB and SVF tests “accounted” for TUG and static balance performances, respectively. I think it is more correct that performance on the FAB and SVF tests impact performances on the TUG and static balance. “Account” would suggest that they explain all of the variance in performances, as assessed via a regression.

Page 8, lines 256-258: Please revise this sentence, as I am not sure what you are trying to express or how the first part of the sentence (age-related alterations) relates to the second part of the sentence (dependency on each cognitive domain). Further, you did not assess how age factored into the effect of the cognitive test performances on the TUG and standing balance dual-task performances.

Page 8, lines 259-260: The phrase “Although the mechanisms explain the distinct influence of cognitive domains on motor tasks” is vague. Please revise to better explain what you’re trying to say.

Page 9, lines 261-262: Did these studies examine smartphone use during a phone call or with text messaging? Please clarify.

Page 9, lines 266-269: Please revise this sentence. It contains some vague language (“greater magnitude of verbal fluency for static balance test”), and I am not sure what you are trying to say with the second clause in that sentence (“but potentially activation…”).

Reviewer 3 Report

The study addresses the potential for cellphone use to have negative motor task consequences by dividing attention. The authors hypothesize the TUG and quiet standing trials would be worse in older adults compared with young adults, and that this effect would be magnified by cell phone use. The manuscript is generally well written but there are a many grammatical errors that need attention. Specific statistical criteria and approach would benefit from revision as well.

General Comments

The authors define higher values as poorer motor performance for the motor tasks, but do not explicitly state whether higher or lower scores are desired on the cognitive tests. In addition to the referenced papers for interpreting the outcomes it would help the reader understand the directionality of those scales as well.

Post hoc statistical power is generally unnecessary. Recommend deleting all references to statistical power. More valuable would be inclusion of the F statistic and degrees of freedom. See:

Zhang, Y., Hedo, R., Rivera, A., Rull, R., Richardson, S., & Tu, X. M. (2019). Post hoc power analysis: Is it an informative and meaningful analysis? General Psychiatry, 32(4), 3–6. https://doi.org/10.1136/gpsych-2019-100069

I question the inclusion of the gender effect. There is no literature support in the Introduction that would suggest differential responses to technology use. Deleting the gender effect would simplify the analyses and reduce the length of the results section, which is a journey to read. This would facilitate emphasis on the critical outcomes that are significant.

Specific Comments

Page 1. Line 31. The reference states the number of cellphone subscriptions. Not users. This does not account for people who own multiple cellphones, one for business, one for personal use, etc. Second, this would not account for shared usage of a single phone, so a user would not be counted as an ‘owner’. Thus, the number of subscriptions does not directly inform the number of users. Obviously, cellphones are ubiquitous, so I appreciate the point but may want to revise this statement give the above.

Page 2. Line 54. ‘In [the] last decade’

Page 2. Line 55. ‘which is [a] consequence’

Page 2. Line 55. Is it that older adults are adapting, or also that a decade has elapsed so people that started using their devices before becoming an older adult have now reached that aging threshold? Probably a combination of new users and ‘aged up’ users. This could have differential effects on their comfortability with the device in general, and while multitasking as is the purpose of this paper.

Page 2. Line 68. Assessments (plural)

Page 2. Line 84. ‘in [the] case’

Page 3. Lines 109-110. Are the variable names provided the ones in the system referenced? These are confusing as Center of Pressure is the metric tracked in order to establish sway range, area, velocity, etc. Can the authors reference traditional definitions such as those provided in the following reference?

Prieto, T. E., Myklebust, J. B., Hoffmann, R. G., Lovett, E. G., & Myklebust, B. M. (1996). Measures of Postural Steadiness: Differences Between Healthy Young and Elderly Adults. IEE Transactions on Biomedical Engineering, 43(9), 956–966.

Page 4. Line 123. What is ‘frontal pocket’? Is this on their pants or on a shirt? Was there variation in clothing tightness that could influence reaching into the pocket and extracting the phone across people?

Page 4. Line 135. ‘The test give[s] insight[s]’. Number agreement

Page 4. Line 147. What effect size do the authors report? What ranges qualify small, medium, or large effects?

Page 4. Line 153. ‘than young [participants] during’

Page 6. Line 188. ‘older group had poor[er] results’

Page 8. Line 235. ‘such as maintain[ing] upright static’ or ‘such as maintain[ance of] upright static’

Page 8. Line 249. ‘standing at [a] bank line’

Page 8. Line 253. No muscle activity (electromyography) information is presented in the present study. Recommend changing this sentence to reflect information that is available (e.g., step counts during the TUG).

Reviewer 4 Report

The study of Lino and Colleagues aims to assess the effects of smartphone cognitive load in motor tasks as TUG and static posture maintenance. The study took into account many young and healthy patients. Some concerns have been highlighted below, particular improvements are needed to understand the methodological steps performed.

Major Concerns

1)Authors stated that they used data acquired from force platforms to carry the analysis. Indeed Center of pressure time course is employed as gold standard data to assess balance maintenance properties of the patients. However, it is not clear which metric were computed from data in order to perform MANOVA analysis.  It is thus required to provide information, since otherwise it is difficult to positively assess the results.

2) Authors have employed cognitive load perturbations to motor tasks assessed. However, this is not new and it is something done also in biomechanical analysis. Hence, Authors should enlarge the literature in the introduction under this perspective. A couple of interesting papers are:
-"Neuromuscular control modelling of human perturbed posture through piecewise affine autoregressive with exogenous input models." Frontiers in bioengineering and biotechnology 9 (2022): 1415.

-"Postural Stability in Young Healthy Subjects - Impact of Reduced Base of Support, Visual Deprivation, Dual Tasking". J. Electromyogr. Kinesiol. 33, 27–33. doi:10.1016/j.jelekin.2017.01.005

Round 2

Reviewer 2 Report

General Comments

The authors have improved the manuscript, but there are still several issues that need to be addressed. Specifically, while the authors state that they have had authors proficient in English and a copyeditor proofread the manuscript, there are still several grammar mistakes throughout the manuscript that need to be corrected, of which I have only identified a few. Also, the authors did not ensure that participants texted without errors, which becomes a confounding factor in their analysis of the dual-task cost on the motor task. Participants who texted without errors may have prioritized the motor task differently than participants who texted with errors. Finally, the authors report a lot of specific results but do not discuss very many of these results. They also do not discuss the implications of the results that they do mention in the discussion section.

Abstract

Page 1, line 21: Because you changed how you were referring to the tests in the methods section of the abstract, you should also change how you refer to them in the results section. Instead of referring to the “TUG and balance test”, you should refer to these tests as the “dynamic and static balance tests” or the “TUG and static balance tests”. You should also keep how you reference these tests consistent throughout the rest of the manuscript text.

Page 1, lines 23 and 28: You are missing an “a” before smartphone in these two lines. The sentences should read “Dual tasking with a smartphone…” and “negative impact of using a smartphone…”.

Introduction

Page 2, lines 54-57: This sentence is still unclear. Please clarify that you are referring to motor and cognitive neural networks. Also, are you stating that the functional connectivity between the two networks is in greater demands or that each individual network is in greater demand when two tasks are performed simultaneously?

Page 2, lines 59-64: You should clarify that the motor performance tends to exhibit greater performance decrements than the cognitive performance during a cognitive-motor dual task. The last sentence still seems disconnected from the rest of the paragraph.

Page 2, line 80: You state that using a smartphone during everyday life is a secondary motor task, but the rest of the introduction seems to suggest that using a smartphone is a cognitive task. Which is it?

Methods

Page 4, line 136: It seems as though you measured “center of pressure sway area” (change in center of pressure over time) not “center of pressure” (the instantaneous location of the center of pressure). Please clarify this in the text.

Page 4, lines 142-143: Please update this sentence to reflect that larger maximum sway in the anterior-posterior and medial-lateral directions, larger center of pressure sway area, and faster imbalance speed were indicative of worse postural balance. Just stating that higher scores indicate worse balance doesn’t indicate to readers what those higher scores would actually look like for each of the variables.

Page 4, lines 148-149: In this sentence you state that the TUG results indicate mobility, but in the abstract you refer to the TUG test as a test of dynamic balance. Please pick one designation and be consistent throughout the manuscript.

 Discussion

Page 12, lines 336-338: This restatement of the study purpose is misleading as it seems to reflect that the secondary purpose (from the secondary hypothesis) was the main aim of the study, as opposed to identifying differences between young and older adults (primary hypothesis).

Page 12, lines 374-383: The authors did not clarify if the previous studies examined smartphone use during a phone call or with text messaging or which motor tasks verbal fluency has been shown to impact. Additionally, the authors do not explain the implications of their results or their results in conjunction with findings from previous literature.

Page 13, lines 400-402: The authors need to better connect this sentence to their results. It is not clear how their results relate to previous studies examining brain activation during a conversation or what implications a relationship to these specific patterns of brain activation may have.

Reviewer 3 Report

No additional comments

Reviewer 4 Report

Authors responded to all my concerns. The paper was consistently revised and it is sutable for publication.